

# Temporal and spatial scale and positional effects on rain erosivity derived from contiguous rain data

Franziska K. Fischer[1,2,3], Tanja Winterrath[4], Karl Auerswald[1]

[1]Lehrstuhl für Grünlandlehre, Technische Universität München, Freising, 85354, Germany
[2]Bayerische Landesanstalt für Landwirtschaft, Freising, 85354, Germany
[3]Außenstelle Weihenstephan, Deutscher Wetterdienst, Freising, 85354, Germany
[4]Zentrale, Deutscher Wetterdienst, Offenbach/ Main, 63067, Germany

*Correspondence to*: Karl Auerswald (auerswald@wzw.tum.de)

**Abstract.** Up until now, erosivity required for soil loss predictions has been mainly estimated from rain gauge data at point scale and then spatially interpolated to erosivity maps. Contiguous radar rain data are now available but they differ in temporal and spatial scale from the point scale. We determined how the intensity threshold has to be modified and which temporal and spatial scaling factors have to be applied to account for the differences in scale. Furthermore, a positional effect

quantifies heterogeneity of erosivity within 1 km², which presently is the highest resolution of freely available gauge-adjusted radar rain data. A method effect accounts for differences in measuring peculiarities between rain gauges and weather radars. These effects were analysed using several large data sets with a total of approximately 2 x 10⁶ erosive events (e.g., records of 115 rain gauges for 16 years distributed across Germany and radar rain data for the same locations and events). With decreasing temporal resolution, peak intensities decreased and the intensity threshold of erosive rains was met

less often. This became especially pronounced, when time increments became larger than 30 min. With decreasing spatial resolution, intensity peaks were also reduced but additionally large areas without erosive rain were included within one pixel. This was due to the steep spatial gradients in erosivity. Erosivity of single events could be zero or more than twice the mean annual sum within a distance of less than 1 km. We conclude that the resulting large positional effect requires use of contiguous rain data, even over distances of less than 1 km, but at the same time contiguously measured radar data cannot be

resolved to point scale. The temporal scale is easier to consider but time increments larger than 30 min should be avoided because the loss of information increases considerably. We provide functions to account for temporal scale (from 1 min to 120 min) and spatial scale (from rain gauge to pixels of 18 km width) that can be applied to rain gauge data of low temporal resolution and to contiguous radar rain data.

## 1 Introduction

Prediction of rain-induced soil erosion using models like the Universal Soil Loss Equation (USLE) requires quantification of the potential of rain to cause soil detachment and transport. This potential is called rainfall erosivity and is typically obtained



from point rainfall measurements using rain gauges. For the conversion of erosivities from point to spatial information, isolines, interpolation techniques and relations to parameters such as the mean summer rainfall depth were used (Rogler and Schwertmann, 1981; Wischmeier, 1959; Wischmeier and Smith, 1958, 1978). The characteristic relation between erosivity and rain depth of the same period was termed erosivity density and used in RUSLE2 (Dabney et al., 2012; USDA, 2013). It

is recommended for areas with poor data availability (Nearing et al., 2017).

Rainfall is now able to be measured contiguously by radars and adjusted by rain gauges so that information about the spatio-temporal distribution of rain is combined with hyetographs measured at ground level. Several countries provide rain-gauge-adjusted radar data products with spatial resolutions of, for example, 1 x 1 km² (Bartels et al., 2004; Fairman et al., 2015), 2 x 2 km² (Koistinen and Michelson, 2002; Michelson et al., 2010), or 4 x 4 km² (Hardegree et al., 2008). Contiguous data of

even coarser scale may result from other sources such as the output of regional climate models (e.g. Christensen et al., 2007; Flato et al., 2013).

Despite the important advantage that radar rain data are contiguous and temporally resolved, they cannot easily be used in place of rain gauge data for erosivity estimations because the scales of measurement differ a lot between both techniques. While rain gauges measure the rain near ground level at point scale (in Germany the collection area is 200 cm²), radars

usually deliver rain measurements with an azimuthal resolution of approx. 1° and a range of 125 m to 1000 m. The data are then typically aggregated in grids of square pixels 1 to 16 km² in size. Rain intensity may differ greatly between point and grid measurements due to reduction in peak intensities with decreasing temporal and spatial resolution. Furthermore, sources of error differ between both measurement techniques. For radar measurements, errors may result from shading of rain cells by objects such as buildings, orographic elevations, or hydrometeors and from the influence of the melting layer causing

bright band effects (Wagner et al., 2012). Major limitations of rain gauges are caused by adhesion, evaporation, wind drift and splashing (Habib et al., 2001). Finally, strong gradients can, in particular, be expected for thunderstorm cells of limited spatial extent. Thus, heterogeneity within pixels will be especially be pronounced for erosive rains (Fiener and Auerswald, 2009; Fischer et al., 2016; Krajewski et al, 2003; Pedersen et al., 2010, Peleg et al., 2016). This heterogeneity cannot be resolved but needs to be quantified because it is the uncertainty that can be expected for predictions at a resolution higher

than the pixel size. This uncertainty also applies in cases where a point measurement of rain erosivity is within a certain distance (e.g. 1 km) from the target area for which erosion is to be calculated. The resulting deviation between point measurement and grid pixel average will be called positional effect in the following. This positional effect should level out in long-term measurements as long as grid pixels are small enough not to include a consistent orographic pattern. It has important implications also for the use of point measurements to predict erosivity and soil loss in the proximity of a

measuring location because it determines the uncertainty, caused by the spatial variability of rain, of these predications.

By definition in the USLE, erosivity is the product of a rain's maximum 30-min intensity and its total kinetic energy (Wischmeier and Smith, 1958). Both factors depend on rain intensity and thus, intensity is squared in erosivity. Consequently, a difference in rain intensity of 10% would already result in difference in erosivity of 21%. Therefore, larger effects of variation in rain intensity can be expected for erosivity than for rainfall. In particular, an average of squares, as



obtained from several point measurements within an area of non-uniform rainfall, will always be higher than the square of the average calculated from the same measurements. This difference between both squares caused by the difference in spatial scale of the measurements is expected to be a robust factor in the long run. We will call this 'spatial scale effect'. A spatial scale effect for erosivity, to the best of our knowledge, has not been studied. This is probably due to the novelty of

operational radar measurements and the lack of long-term data sets required for erosivity estimations. Long-term and revised radar rain data now exist and can help to improve contiguous erosivity and soil loss estimations. Therefore, it is crucial to know to which extent erosivity, and subsequently also soil loss, are underestimated due to the spatial scale effect by gridded rain data as provided by radar measurements and also by climate models that employ an even coarser spatial resolution than typical radars (Chen and Knutson, 2008). Rain intensities from radar may additionally be smoothed by measuring and

subsequent processing procedures. The contribution of erosivity underestimation due to these procedures is called 'method effect' in the following. Thus, the difference in erosivity from rain gauge data and from radar data is caused by spatial scale and method effects.

Another effect is induced by the temporal scale of the data used for erosivity calculations. With decreasing temporal resolution, mainly maximum 30-min intensity, and hence erosivity, are increasingly underestimated. Therefore, temporal

scaling factors are required to compensate for this underestimation (e.g. Auerswald et al., 2015; Agnese et al., 2006; Istok et al., 1986; Williams and Sheridan, 1991; Weiss, 1964; Yin et al., 2007). These are especially important for contiguous data, for which temporal resolution of rain data is decreased, often to 60 min, as a requirement for the adjustment to rain gauge data and to reduce the enormous amount of data caused by the high spatial resolution and wide spatial and temporal coverage.

We therefore hypothesize that: 1) with decreasing temporal and spatial resolution of rain data, calculated erosivities decrease due to a smoothing of intensities; 2) radar measurements cause an additional underestimation of erosivities due to the measuring principle and the required calculation and correction steps; 3) large uncertainty of erosivity within 1 km² is due to the positional effect. The effects of hypotheses 1) and 2) have to be compensated by changes in the calculation of erosivity while the effect of hypothesis 3) quantifies uncertainty of erosivity of individual events at any location within an area of 1

km² around a rain gauge.

## 2 Material and methods

### 2.1 Erosivity calculation procedures

Following Wischmeier (1959) and Wischmeier and Smith (1978) erosivity of a single rain event ($R_e$) was calculated as the product of the maximum 30-min rain intensity ($I_{max30}$) and the kinetic energy ($E_{kin}$) (Eq. (1)). A rain event is erosive by

definition if it has a total precipitation ($P$) of at least 12.7 mm or an $I_{max30}$ of 12.7 mm h$^{-1}$ (min($I_{max30}$)).

$$R_e = I_{max30} * E_{kin} \tag{1}$$





The $E_{kin,i}$ per mm rain depth (in kJ m$^{-2}$ mm$^{-1}$) was calculated for intervals $i$ of constant rain intensity $I$ following Eqs. (2.1) – (2.3). For all intervals $i$, $E_{kin,i}$ was multiplied with the rain amount of this interval and then summed up to yield $E_{kin}$ for the entire event.

$$E_{kin,i} = (11.89 + 8.73 * log_{10}I) * 10^{-3} \qquad \text{for } 0.05 \text{ mm h}^{-1} \leq I < 76.2 \text{ mm h}^{-1} \qquad (2.1)$$

$$E_{kin,i} = 0 \qquad \text{for } I < 0.05 \text{ mm h}^{-1} \qquad (2.2)$$

$$E_{kin,i} = 28.33 * 10^{-3} \qquad \text{for } I \geq 76.2 \text{ mm h}^{-1} \qquad (2.3)$$

When $I_{max30}$ was derived from data with intervals longer than 30 min, $I_{max30}$ was determined as the maximum rain intensity of the event. Erosive events are separated from each other by rain breaks of at least 6 hours (Wischmeier and Smith, 1958, 1978). For example, using 60 min rain data, we defined events as being separate when five subsequent 60-min intervals without rain occurred. This assumes that rain events stop and start on average in the middle of the first and the last non-zero rain interval. The same concept was used for all data sets with temporal resolutions > 60 min.

The annual erosivity of a specific year ($R_y$) is the sum of $R_e$ of all $n$ erosive events within this year. The long-term average annual erosivity ($R$) is then calculated as

$$R = \frac{1}{k} \sum_{j}^{k} \left( \sum_{i}^{n} R_{e,i} \right)_j = \frac{1}{k} \sum_{j}^{k} R_{y,j} \qquad (3)$$

which is the average of $R_y$ for a number of $k$ years, in case of this study 16 years.

While in the USA and other countries often the unit MJ mm ha$^{-1}$ h$^{-1}$ is used, we use N h$^{-1}$ for $R_e$, because it is the unit for $R$ most often used in Europe and because of its simplicity. The units can be easily converted by multiplying the values in N h$^{-1}$ with a factor of 10 to yield MJ mm ha$^{-1}$ h$^{-1}$.

**2.2 Determination of scale effects**

The smoothing caused by decreasing resolution in time and space mainly decreases intensity, while the total amount of rainfall should, in principle, be unaffected. This decrease in intensity has two consequences. First, the intensity threshold $min(I_{max30})$ that defines an erosive event is less often met and thus has to be adjusted to arrive at the same number of erosive rains irrespective of resolution. Second, scaling factors for $R_e$ are required. A temporal scaling factor $t_{\tau,\sigma}$ scales from temporal resolution $\tau$ to 1-min resolution at a certain spatial scale with pixel width $\sigma$. A spatial scaling factor $s_\sigma$ scales from spatial resolution $\sigma$ to point resolution (rain gauge). A method effect $m$ may additionally occur, which quantifies the difference between erosivities obtained from rain gauges and from radar measurements at identical spatial and temporal scales. It is caused by the additional smoothing resulting from the radar technique and the adjustment and correction steps subsequently required. It may also include the errors of rain measurement that differ between the methods rain gauge and radar. The positional effect $p_{Re}$ describes the average relative deviation of erosivity of single events derived at 1 km resolution and at point scale from rain gauges located within the respective 1 km pixel including the spatial scale and method effects. The positional effect cannot be used for correction but it is a measure of variability within a certain pixel.



Adjusting the intensity threshold to account for smoothing at low resolution is appropriate only for the temporal resolution. With decreasing spatial resolution some areas will be included within a pixel that actually received erosive rain, while other areas within the pixel did not. Without adjustment of the intensity threshold the entire pixel may be classified as non-erosive, while adjustment of the threshold would then indicate an erosive event also in those areas within a pixel where no erosive

rain had occurred. Adjusting the intensity threshold with decreasing spatial resolution could not correct both errors simultaneously.  Even more important, the criterion of breaks that separate between events is biased for large areas. Any rain at some place within a large pixel abrogates an existing break even if it does not fall at a site that experienced an erosive rain. The loss of a break with increasing pixel size decreases the number of events even when all events are considered. Adjusting the number of events in this case would be a wrong correction. Hence for the spatial resolution the threshold effect was

included in $s_\sigma$, while for the temporal scale effect the intensity threshold could be adjusted. As a result the number of erosive events can correctly be estimated at low temporal resolution with this adjustment at point scale while for a spatial resolution lower than point scale the number of erosive events will be wrong compared to point scale. Only the sum of erosivities over a longer period of time (months, years, or longer) can then be corrected with the spatial scaling factor.

To cover a wide range of spatial and temporal resolutions, several large and overlapping data sets had to be combined (for an

overview see Table 1). The spatial resolution from point scale to 1 km pixel width (with an intermediate pixel width of 0.5 km) was covered by a high-density network of 12 rain gauges operated over four years within an area of 1 km² (taken from Fiener and Auerswald 2009; for location of the measuring site see Fig. 1a; for the spatial distribution of rain gauges see Fig. 1c). It included 542 events at point scale. The average deviation of annual erosivities calculated from hyetographs at point scale and from spatially integrated hyetographs at 0.5 km or 1 km pixel width (here referred to as 'pseudo-radar' data)

yielded the spatial scaling factors $s_{\sigma=0.5}$ and $s_{\sigma=1}$. The individual deviation of event erosivities at point scale from the average was due to the positional effect $p_{Re}$ (for an example see Fig. 1b). The average positional effect $p_{Re}$ was calculated as the geometric mean of the $k$ ratios of $R_e$ derived from rain gauge ($\sigma = 0$) and 1 km² pixel data ($\sigma = 1$), for which neither rain gauge $R_e$ nor pixel $R_e$ was zero:

$$p_{Re} = 10^\wedge(\textstyle\sum_{i=1}^{k} log_{10}(R_{e,\sigma=0}/R_{e,\sigma=1})_i / k). \hspace{2cm} (4)$$

The positional effects were determined separately for events with $R_{e,\sigma=1}$ larger and $R_{e,\sigma=1}$ lower than $R_{e,\sigma=0}$. Rains that were erosive at only one of both spatial scales were excluded from the calculation of the geometric mean and the percentages of these events were determined for both cases.

Point scale and 1 km pixel width was also compared for a much wider data set covering the whole of Germany by comparing erosivities at 115 rain gauges with erosivities obtained from radar data with 1 km resolution (for location of the rain gauges

and the coverage of weather radars see Fig. 1a). The 115 radar pixels in which the rain gauges were operated, were selected. Rain gauge data were taken from the Climate Data Center of the German Weather Service (ftp://ftp-cdc.dwd.de/pub/CDC/). The German Weather Service also provided the radar data, which were a revised version of the radar rain data product RADOLAN (Winterrath et al., 2012; Winterrath et al., 2017). This resulted in point-pixel pairs for >20,000 erosive rain





events. The long-term (16 years) average deviation of $R$ between point and pixel scale was due to the smoothing effects of the spatial scale effect and the radar technique (method effect). The method effect was quantified by subtracting the spatial scale effect, as obtained from the dense rain gauge network, from the combined effect, as obtained by comparing erosivities from rain gauges with radar-derived erosivities. The combined effects of spatial scale and method were also tested for

seasonal variation.

For spatial resolution lower than 1 km pixel width, radar data were aggregated to yield pixel widths of up to 18 km. Erosivities were calculated from the aggregated rain data and compared to the erosivities at 1 km pixel width, which were averaged for the pixel width being examined. This comparison was carried out for radar data covering an area of 800 x 600 km² over 2 months, which comprised $1.9 \times 10^6$ events at 1 km pixel width (Table 1).

The temporal resolutions of the rain gauge data and the radar data differed (1 min, 5 min, 1 h). Erosivities derived from these data were adjusted to 1 min resolution with the appropriate temporal scaling factor. The temporal scaling factors were determined on two spatial scales, at point scale and at 1 km pixel width. To this end, 17 out of the 115 point-pixel pairs were selected randomly and rain data for the period 2001 to 2016 (16 years) with 1 min resolution from rain gauges and 5 min resolution from radar measurements were used. The rain gauge data yielded a total of 4,599 erosive events, for which rain

data were aggregated to 2 min, 5 min, 10 min, 15 min, 30 min, 45 min, 60 min, 80 min, 100 min and 120 min intervals, and $R_e$ was determined as described in Sect. 2.1. The intensity threshold $\min(I_{\max30})_\tau$ was adjusted until the annual number of erosive rain events at the respective temporal resolution $\tau$ was equal to that at $\tau = 1$ min. The temporal scaling factor ($t_{\tau=x,\sigma=y}$) for $R_e$ was then obtained at point scale ($\sigma = 0$) from:

$$t_{\tau=x,\sigma=0} = \sum_{i=1}^{N}(R_{e,\tau=1,\sigma=0})_i \ / \ \sum_{i=1}^{N}(R_{e,\tau=x,\sigma=0})_i \tag{5}$$

which is the ratio of the sums of $R_e$ derived from 1 min data and $R_e$ derived from data with $\tau > 1$ min at point scale. Additionally, for 1 km pixel width $t_{\tau=x,\sigma=1}$ was estimated by using an intermediate radar product of RADOLAN with a temporal resolution of 5 min that was recursively adjusted corresponding to the 60 min RADOLAN data (analogously to Fischer et al. 2016). This was done for the 17 grid pixels where the 17 rain gauges were located. The temporal scaling factors were derived from RADOLAN data as described above (Eq. (5)) but relative to $\tau = 5$ min. The resulting factors were then

multiplied by the scaling factor for $\tau = 5$ min obtained from the rain gauge data to yield scaling factors relative to a temporal resolution $\tau = 1$ min.

The temporal scaling factors $t_{\tau=x,\sigma=0}$ were additionally determined for each month (Jan – Dec) and separately for rain gauges located in the northern and southern halves of Germany (7 and 10 rain gauges, respectively) to test for any seasonal or regional dependence of the factors.

Finally, the combined procedure of an adjusted intensity threshold and a temporal scaling factor was validated by comparing annual $R_y$ obtained from 60-min RADOLAN data to $R_y$ derived from RADOLAN data with 5 min resolution. This was done for the remaining 98 (115 – 17) grid pixels and 16 years yielding a total of 1568 $R_y$.





## 2.3 Statistics

We mainly used arithmetic means even though most distributions were strongly skewed. Arithmetic means are less robust than other measures like geometric means but our huge sample size compensated for this. Using arithmetic means instead of robust measures is a requirement of the USLE, which sums up erosivities over one year or longer. The arithmetic mean

provides an unbiased estimator of event erosivity that allows sums to be calculated over longer periods of time (e.g. one year). Otherwise different scaling factors would become necessary for individual events and for temporal sums depending on their temporal length.

Statistical spread is quantified by the standard deviation (SD) or the root mean squared error (RMSE), and the uncertainty of the scaling factors is quantified by their 95% interval of confidence (CI). Validation included the calculation of the Nash–

Sutcliffe efficiency (Nash and Sutcliffe, 1970).

## 3 Results

### 3.1 Temporal scale effect

With 17 rain gauges operating at 1 min resolution, 4599 erosive events were determined in 16 years. $R_e$ ranged from 0.1 N h$^{-1}$ to 178.4 N h$^{-1}$ with an average of 5.8 N h$^{-1}$. The number of events with $P \geq 12.7$ mm or $I_{max30} > 12.7$ mm h$^{-1}$ decreased

pronouncedly when resolution decreased from 1 min down to 120 min (by 1%, 14% and 16% at a resolution of 2 min, 60 min and 120 min, respectively). To avoid this loss of events, $\min(I_{max30})_\tau$ was decreased continuously with decreasing temporal resolution (Fig. 2b). The decrease was less steep below a temporal resolution of 30 min than above:

$$\min(I_{max30})_\tau = -0.59\,\tau^{0.5} + 13.23 \qquad \text{for } \tau \leq 30 \text{ min} \qquad\qquad (6.1)$$

$$\min(I_{max30})_\tau = 147\,\tau^{-0.79} \qquad\qquad \text{for } \tau \geq 30 \text{ min} \qquad\qquad (6.2)$$

This change at a resolution of 30 min is because 30 min is the time interval in which the maximum is searched for. For resolutions higher than 30 min, there is a discrepancy between the true period of $I_{max30}$ and the period of $I_{max30}$ that is coerced by the temporal resolution (see grey bars in Fig. 2a). The error caused by this discrepancy only results from the difference in intensity immediately before and after true $I_{max30}$. When the temporal resolution becomes less than 30 min, attenuation caused by the period exceeding the 30-min interval additionally decreases intensity (see 60-min resolution in Fig. 2a). This

attenuation increases the lower the temporal resolution becomes, and caused Eq. (6.2) to be much steeper than Eq. (6.1).

The decrease in $\min(I_{max30})_\tau$ was identical for both, the rain gauge scale and the 1 km² scale (slope between both scales: 1.0067, r² = 0.9858, n = 9). For both scales combined, RMSE was only 0.10 and 0.39 for Eqs. (6.1) and (6.2) respectively. Thus, both equations were valid for point scale and for a grid width of 1 km.

Rain erosivity also decreased with decreasing temporal resolution and, in turn, the scaling factor $t_{\tau,\sigma}$ increased (Fig. 2c; Eqs. (7.1) – (7.2b)). For intervals $\tau \leq 30$ min, the increase was identical for rain gauge scale and for radar pixels of 1 km pixel





width. The increase of $t_{\tau,\sigma}$ was much steeper when $\tau$ became longer than 30 min. This increase then depended on the spatial scale and was larger for rain gauge scale than for radar pixels of 1 km pixel width (Fig. 2c). The behaviour of $t_{\tau,\sigma}$ was caused by underestimating $E_{kin}$ and underestimating $I_{max30}$. The underestimation of $I_{max30}$ was the stronger effect (data not shown). It prevailed for time intervals greater than 30 min and caused the break at a temporal resolution of 30 min, as already shown

for $\min(I_{max30})_\tau$. The identical behaviour of intensity with decreasing temporal resolution at rain gauge scale and at 1 km² radar pixel scale that was already evident for $\min(I_{max30})_\tau$ thus also led to identical $t_{\tau,\sigma}$ for both spatial scales as long as $\tau$ was less than 30 min. For $\tau > 30$ min the attenuation of intensity peaks came into play. This attenuation was less for the 1 km radar data than for the rain gauge data because the time a moving intensity peak remains in a 1 km² grid pixel is longer than the time it requires to pass a rain gauge. In consequence, three equations for $t_{\tau,\sigma}$ (Eqs. (7.1) – (7.2b)) were necessary to adjust

$R_e$ , $R_y$ or $R$ to 1 min resolution at the respective spatial scale.

$$t_{\tau,\sigma} = \frac{\tau}{100} + 1 \qquad \text{for } \tau \le 30 \text{ min and point or 1 x 1 km² grid scale} \qquad (7.1)$$

$$t_{\tau,\sigma=0} = \frac{\tau}{40} + 0.55 \qquad \text{for } \tau \ge 30 \text{ min and point scale} \qquad \text{or} \qquad (7.2a)$$

$$t_{\tau,\sigma=1} = \frac{\tau}{50} + 0.70 \qquad \text{for } \tau \ge 30 \text{ min and 1 x 1 km² grid scale.} \qquad (7.2b)$$

The RMSE of all three equations was less than 0.04. The validity of combining the effects of $\min(I_{max30})_{\tau=60}$ and $t_{\tau=60,\sigma=1}$ was

supported by the close correlation of temporally scaled $R_y$ derived from 5 min and 60 min RADOLAN data, for which the Nash-Sutcliffe efficiency was 0.9483 (n = 1568) while RMSE was 8.8 N $h^{-1}$ $yr^{-1}$.

Variation among monthly $t_{\tau,\sigma=0}$ was small, especially for $\tau \le 60$ min. The coefficient of variation among monthly $t_{\tau,\sigma=0}$ was $\le 6\%$ for $\tau \le 60$ min and 11% to 14% for $\tau > 60$ min. It was not clear if there was seasonality in this variation because for some temporal resolutions $t_{\tau,\sigma=0}$ was higher for summer than for winter months, while for other resolutions the opposite was

the case.

There was also a negligible regional variation for $\tau > 30$ min, while no difference could be found for $\tau \le 30$ min. For intervals longer than 30 min the scaling factor $t_{\tau,\sigma=0}$ increased slightly more in northern Germany (+4%) than in southern Germany (-2%), compared to the whole of Germany. This small difference will only become relevant if data of very low temporal resolution are used.

**3.2 Spatial scale effects**

Erosivities of all data of rain gauge-radar pixel pairs were calculated by application of appropriate $\min(I_{max30})_\tau$ and temporal scaling factors to enable comparison. Annual erosivity $R_y$ for the 0.5 x 0.5 km² pseudo-radar data set was 7.3% lower than the average of the rain gauges. This resulted in a factor $s_{\sigma=0.5}$ of 1.08 (CI: 1.00 – 1.16). This factor increased to $s_{\sigma=1} = 1.15$ (CI: 1.04 – 1.26) if $R_y$ was calculated from 1 x 1 km² pseudo-radar data (Fig. 3).

For the rain gauges of the 115 rain gauge-radar pixel pairs, long-term annual $R$ varied between 42 and 223 N $h^{-1}$ $yr^{-1}$ over 16 years and was on average 90.2 N $h^{-1}$ $yr^{-1}$. For the radar pixels, $R$ varied between 26 and 146 N $h^{-1}$ $yr^{-1}$ but was on average


only 62 N h$^{-1}$ yr$^{-1}$ (Fig. 4). In this case the deviation was equal to a factor of 1.48 (CI: 1.43 – 1.52), which was considerably larger than $s_{\sigma=1}$ obtained from pseudo-radar data, for which no difference in measurement method occurred between point scale and pixel scale. This difference was hence assigned to a method effect (Fig. 3).

The monthly comparison of the 115 rain gauge-radar pixel pairs over 16 years did not yield significant differences between
months due to the large CI of the combined scale and method effects (CI between ±4% to ±9% for the individual months) but on average this combined effect was lower during the hydrological winter months (1.16; CI: 1.12 – 1.21) than during the hydrological summer months (1.42; CI: 1.30 – 1.53). This difference, despite being significant ($p < 0.001$), was unimportant because of the small contribution of winter months to annual erosivity.

For the large and contiguous radar data set of 800 x 600 pixels, 1.9 x 10$^6$ events were recorded at 1 x 1 km² scale. For these
events, $R_e$ was on average 5.1 N h$^{-1}$ and ranged from 0.5 to 1270 N h$^{-1}$. Aggregating these pixels to larger square pixels decreased $R_e$. At 18 x 18 km², $R_e$ was on average 4.4 N h$^{-1}$ and ranged from 0.2 to 221.6 N h$^{-1}$. In consequence, the spatial scaling factor $s_\sigma$ increased further (Fig. 3). The increase in scaling factors over the entire range from point scale to 18 km grid width could be described by a multiple regression ($r^2 = 0.9995$, n = 21) accounting for pixel width $\sigma$ (in km) and the method effect $m$ depending on the method $\mu$ (which is 0 for rain gauges and 1 for radar data):

$$m + s_\sigma = 1 + 0.35\,\mu + 0.092\,\sigma^{3/4} \tag{8}$$

The CI was $\pm\,0.004$ for the slope of $\sigma$ and $\pm\,0.02$ for the method effect.

On average for the pseudo-radar pixel, rain was erosive for only 10 out of 12 rain gauges. Hence only 83% of the 1 km² pixel was covered by an erosive event. The fraction covered by the erosive event decreased further the larger the pixel size became
(fraction = 83% - 10.3 × ln(pixel size (km²)), $r^2 = 0.9974$, n = 18). On average only about 50% of a 5 × 5 km² pixel and 25% of a 17 × 17 km² pixel received an erosive rain. This makes it increasingly difficult to detect erosive rains the larger pixel size becomes, which caused the strong increase in the spatial scaling factor and indicated a strong positional effect.

### 3.3 Positional effects

The positional effect as defined here describes the variability of $R_e$ within 1 x 1 km². Using the pairs with the true radar data, 29 610 erosive rain events were recorded during 16 years at the 115 rain gauges. On average, $R_e$ was 5.6 N h$^{-1}$ and ranged from 0.1 to 547.2 N h$^{-1}$. For the corresponding 115 radar pixels, 25 884 erosive events were recorded during the 16 years. Mean $R_e$ was 4.4 N h$^{-1}$ and ranged from 0.2 to 318.9 N h$^{-1}$.

Combining all events of the 115 rain gauge-radar pixel pairs during 16 years that were at least erosive at rain gauge scale or
at radar pixel scale resulted in 35 124 events. Only 57% of them were erosive at both scales, while the criteria for an erosive event were met exclusively at pixel scale for 16% of all events and exclusively for 27% of all events at rain gauge scale (Table 2). The gradients of erosivity within 1 km² were huge. The largest event that was recorded at a rain gauge while the radar pixel indicated no erosive event was 156 N h$^{-1}$. The largest event for the opposite case, i.e. that radar recorded an



erosive event while the rain gauge recorded no erosive event, was similarly high (180 N h$^{-1}$). The mean $R_e$ of erosive events, which were recorded for the radar pixel while $R_e$ at the corresponding rain gauge was zero, was 2.9 N h$^{-1}$ (SD: $\pm$ 4.9 N h$^{-1}$). The mean $R_e$ of events, which were erosive at a rain gauge but not for the corresponding radar pixel, was also 2.9 N h$^{-1}$ (SD: $\pm$ 5.6 N h$^{-1}$).

The percentage of unpaired events was not significantly related to the geographical location, neither longitude (r = -0.02, $p$ = 0.23) nor latitude (r = -0.01, $p$ = 0.83). It was also independent of the distance to the adjacent radar station (r = -0.02, $p$ = 0.79), which might be used as proxy for increasing noise in the radar data. The percentage was higher in winter (Oct – Mar) with 34% (SD: $\pm$ 2.4%) than in summer (Apr – Sept) with 25% (SD: $\pm$ 2.4%). The probability of remaining just below the threshold of an erosive event on one of both scales was higher in winter than in summer as in general winter events are less

intensive than summer events. Mean $R_e$ in winter was only 35% of mean $R_e$ in summer.

Rain gauge $R_e$ was larger than radar $R_e$ for 74% of those point-pixel pairs (points above the line of unity in Fig. 5) which were erosive on both scales (19 944 events). Mean $p_{Re}$ was 1.54 (CI: $\pm$ 0.01) for these events. This value quantifies the mean deviation of all locations within a 1 km² pixel that experience a higher erosivity than the mean. For individual locations, the deviation can be much larger, which was already evident from the magnitude of the largest events that were recorded only on

one of both scales. For individual locations with an erosive event on both scales, $p_{Re}$ could be considerably higher than 10 (see "outliers" in Fig. 5). Rain gauge $R_e$ was lower than radar $R_e$ for only 26% of all events (points below the line of unity in Fig. 5) and $p_{Re}$ was 0.72 (CI: $\pm$ 0.01). Again, the deviation of individual locations within 1 km² could be much larger.

For the dense rain gauge field used to create pseudo-radar data, 579 point-pixel pairs of events were at least erosive at rain gauge scale or at pseudo-radar pixel scale. For these 579 events, $R_e$ derived from rain gauge data ranged from 0 to 45.5 N h$^{-1}$

(mean 3.9 N h$^{-1}$) and $R_e$ derived from pseudo-radar data ranged from 0 to 28.1 N h$^{-1}$ (mean 3.4 N h$^{-1}$) (Fig. 6). For 9% of these events, the event was not erosive with pseudo-radar but at the rain gauge and for 6% the opposite was true (Table 3).

For 67% of those events which were erosive at both scales, rain gauge $R_e$ was larger than pseudo-radar $R_e$ and $p_{Re}$ was 1.28 (CI: 1.25 – 1.30). For 33% of these events, rain gauge $R_e$ was lower than pseudo-radar $R_e$ and $p_{Re}$ was 0.81 (CI: 0.77 – 0.85). Also in this case, where measurement errors could be excluded because rain gauge $R_e$ and pseudo-radar $R_e$ were calculated

from the same data, the variation within 1 km² was again huge. For the single days with erosive events, $R_e$ varied greatly between rain gauges. For an example see height of the rectangle in Fig. 6. Although this was the largest event in this data set, one rain gauge remained below the threshold and hence recorded no erosive event. This large variation was also reflected by the large coefficient of variation between rain gauge $R_e$ for the same day (mean 68%).

## 4 Discussion

Our analysis showed pronounced effects of temporal scale, spatial scale, position and measuring method. These effects were all caused by the sensitivity of erosivity calculation to intensity peaks and because thresholds were used for the definition of erosivity. These strong effects call for using temporally and spatially highly resolved rain gauge measurements, like those





used in the development of the USLE and most subsequent studies. Our study, however, also showed strong gradients in erosivity that were also caused by the sensitivity to intensity peaks and by the thresholds, which earlier studies also showed (Fiener and Auerswald, 2009; Fischer et al., 2016; Krajewski et al, 2003; Pedersen et al., 2010, Peleg et al., 2016). Erosivity can thus reliably be recorded at the position of a rain gauge but this information cannot even be extrapolated over a distance

of only 500 m (half of our radar pixel widths). This was illustrated by the fact that within this distance, $R_e$ could be zero or $>150$ N h$^{-1}$, which is more than twice the annual erosivity in Germany (Auerswald 2006, Sauerborn 1994). It is also illustrated by the fact that the largest $R_e$ that was recorded within only two months was 1270 N h$^{-1}$ when contiguous measurements were used, while the largest $R_e$ that occurred during 16 years when the same region was covered by 115 rain gauges was only 547 N h$^{-1}$. Hence rain gauge measurements fail to record many erosive events that occur in their close

vicinity (even < 500 m). Erosivity measured at a rain gauge cannot be extrapolated to a small watershed, to farms or even to fields. Discrepancies between model predictions and measurements of erosion that can be found in many studies (Govers 1991, Liu et al. 1997, Risse et al. 1993, Rüttimann et al. 1995, Zhang et al. 1996) probably originate in part from this strong positional effect. Such strong discrepancies during individual events even exist between replicates of bare plots (Nearing et al. 1999) or between replicated vegetated plots and cannot be explained by plot characteristics for subsequent runoff and soil

loss observations (Wendt et al. 1986). Erosion prediction and model development is thus strongly limited by the unexplained variability caused by short-range erosivity gradients. Hence, there is no alternative to using contiguous rain measurements. Radar technology provides, for the first time, measurements that fulfil this need.

Contiguous measurements, on the other hand, suffer from the fact that they cannot be carried out at the same temporal and spatial scale as rain gauge measurements, and the method of measurement differs. Here we provide scaling factors that help

to partly overcome this problem and which allow radar measurements to be used for erosivity calculations. These factors, however, do not solve the problem that contiguous measurements integrate over a certain space and time and thus the information about the variation within these domains is lost. In particular, the positional effect can only be used to quantify uncertainty within a radar pixel but it cannot be used to predict erosivity at specific locations within a pixel. This large uncertainty is probably also one of the main reasons for the discrepancy between observed soil loss and predicted soil loss

based on radar rain data for individual fields, whereas this discrepancy disappeared as soon as many fields were grouped, irrespective of how this grouping was done (Fischer et al., 2018, Auerswald et al., 2018). With future improvements in technology it may become possible to further improve temporal and spatial resolution of contiguous rain data and, thus, to reduce the uncertainty of event erosivities.

Temporal scaling factors had already been developed (Auerswald et al., 2015; Agnese et al., 2006; Istok et al., 1986;

Williams and Sheridan, 1991; Weiss, 1964; Yin et al., 2007) because they are also required for rain gauge measurements of low temporal resolution (in data storage). Our temporal scaling factors were of a similar order of magnitude to those in other studies. However, our data showed that using a scaling factor is not sufficient because the intensity threshold also has to be adjusted in order to identify the correct number of erosive events. The existence of an erosive event and long-term sums of erosivity will otherwise be incorrect, even with a temporal scaling factor. To our knowledge our study provides, for the first





time, a function that enables the intensity threshold to be adjusted according to the temporal resolution of the rain data. Adjustment of the total rain depth threshold is not necessary because total rain depth should be independent of the temporal resolution, as long as it is still short enough to identify the rain breaks that separate individual events.

Despite providing intensity thresholds and scaling factors for $R_e$, $R_y$ and $R$ for different temporal resolutions, we advocate for
using a high resolution in order to not lose information. All scaling factors can only represent average behaviour and cannot reflect the behaviour of an individual event. A high resolution is easier to achieve in the time domain than in the spatial domain. In particular, it is advantageous to have a temporal resolution that is higher than 30 min because scaling factors increased strongly for less resolved data. For shorter time increments, only compensation for the error that resulted from an imperfect identification of the period of $I_{max30}$ was necessary. Longer time increments than 30 min additionally attenuated
$I_{max30}$ and thus blurred this information.

The spatial scale was more difficult to consider than the temporal scale due to the large positional effect. In particular, large parts of a pixel remained below the thresholds of an erosive event even when measurement errors could be excluded, like in the case of the pseudo-radar pixel that used rain gauge measurements. On average, 17% of the rain gauges within a 1 km² pixel remained below the erosivity threshold while the other rain gauges recorded an erosive event. This percentage
increased strongly with increasing pixel size. In consequence, the spatial scale effect cannot be corrected for individual events but only for the averages of many events.

The spatial scaling factor is conceptually the inverse of the so-called areal reduction factors, which are used to reduce rain intensity from rain gauge measurements when scaled to catchment areas depending on the duration and return period of the rain event (Allen and DeGaetano, 2005; De Michele et al., 2001; Stewart, 1989). This conceptual difference is due to the
difference in the intended purpose of contiguous rain data. While in catchment hydrology the average and the relative distribution of rain depth within a watershed is of interest (Asquith and Famiglietti, 2000), for erosion analysis, rain intensities are important at point and field-scale where erosion occurs.

The method effect combines all differences in measurement and measuring errors (e.g., the wind effect in the case of rain gauges). It is thus highly dependent on the specific configuration of rain gauge measurements and radar measurements,
including all subsequent data manipulation steps. These configurations are usually fairly standardized within a country (e.g., rain gauge height and diameter are usually defined) but differ from country to country. Our method effect may thus only be valid for Germany and application to other countries, even if they use similar rain gauge and radar protocols (e.g., Goudenhoofdt and Delobbe, 2016; Koistinen and Michelson, 2002), should be done with care.

As an example, for the new German RADOLAN product that soon will become publicly available (spatial resolution 1 km²,
temporal resolution 60 min) the $I_{max30}$ threshold would have to be lowered to 5.79 mm h$^{-1}$ while the total precipitation threshold remains at 12.7 mm. The temporal scaling factor would be $t = 1.9$, the spatial scaling factor would be $s = 1.13$ to which the method effect of $m = 0.35$ has to be added. In total, the correction factor would be 2.81 (($1.13 + 0.35$) × 1.9). Hence the change of the $I_{max30}$ threshold and the combined scaling factor are large and ignoring both would considerably





underestimate erosivity. The large change of the $I_{max30}$ threshold and the large temporal scaling factor also show that much information is lost when using data of 60 min resolution.

This loss of information can either be an advantage or a disadvantage. It would be a disadvantage in hindcasting when usually the true pattern of erosivity is wanted. In this case a better resolved product like 5-min data should be used. The $I_{max30}$

threshold would then be 11.9 mm h$^{-1}$ and the temporal scaling factor would only be t = 1.05, indicating a minor loss of information. The spatial scaling factor is already rather low and the method effect cannot be avoided.

On the other hand the loss of information would be an advantage in forecasting, which aims at the likely regional pattern of erosivity. The loss of information removes the influence of randomly occurring local events of extraordinarily high magnitude that add noise to the regional pattern of erosivity. The finding that the largest $R_e$ within only two months was

1270 N h$^{-1}$ while the expected long-term average $R$ was only about 70 N h$^{-1}$ yr$^{-1}$ (Sauerborn 1994) shows that this single event would add 64 N h$^{-1}$ yr$^{-1}$ to a 20-yr record of radar data. Even in a 100-yr record this single event would still be detectable. Using data of 60 min resolution thus reduces the need for smoothing the map statistically to remove the influence of such local events.

**5. Conclusions**

Large gradients in event erosivity occur that can only be captured by contiguous rain data. Radar technology enables such contiguous rain data to be recorded but not at the same temporal and spatial scale as measurements from rain gauges. Using data of lower temporal and spatial resolution than rain gauges leads to a pronounced underestimation of erosivity. Here we provide a set of correction functions that enable this underestimation to be corrected. In particular, the intensity threshold has to be modified, a temporal scale factor, a spatial scale factor and a factor accounting for measurement peculiarities have to be

considered. In combination with contiguous radar rain data this could be a major step forward in erosion modeling.

Author contribution

KA and FF designed the analysis, which was mainly carried out by FF. TW provided most data and the knowledge about all steps involved in radar data creation. FF and KA prepared the manuscript with contributions by TW.

**Acknowledgements**

This study was part of the project "Ermittlung des Raum- und Jahreszeitmusters der Regenerosivität in Bayern aus radargestützten Niederschlagsdaten zur Verbesserung der Erosionsprognose mit der Allgemeinen Bodenabtragsgleichung" at the Bavarian State Research Center for Agriculture (PI Robert Brandhuber) and funded by the Bayerisches Staatsministerium für Ernährung, Landwirtschaft und Forsten (A/15/17). Karin Levin provided language editing.



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





**Figure 1:** a) Locations of the 115 rain gauges (dots), the coverage (circles) of the 17 weather radars (crosses) and the location of the 12 rain gauges used for the pseudo-radar data (square; size exaggerated) in Germany. b) One rain gauge (dot) within one 1 x 1 km² pixel (bounding box) and isolines of rain depth (taken from Fiener and Auerswald, 2009) illustrating the variability of a single erosive rain event at 1 x 1 km² grid scale causing positional effects. c) Distribution of the 12 rain gauges (dots) within an area of 1 x 1 km² (bounding box) and their corresponding Thiessen polygons. Dashed lines separate the area to a spatial scale of 0.5 x 0.5 km².


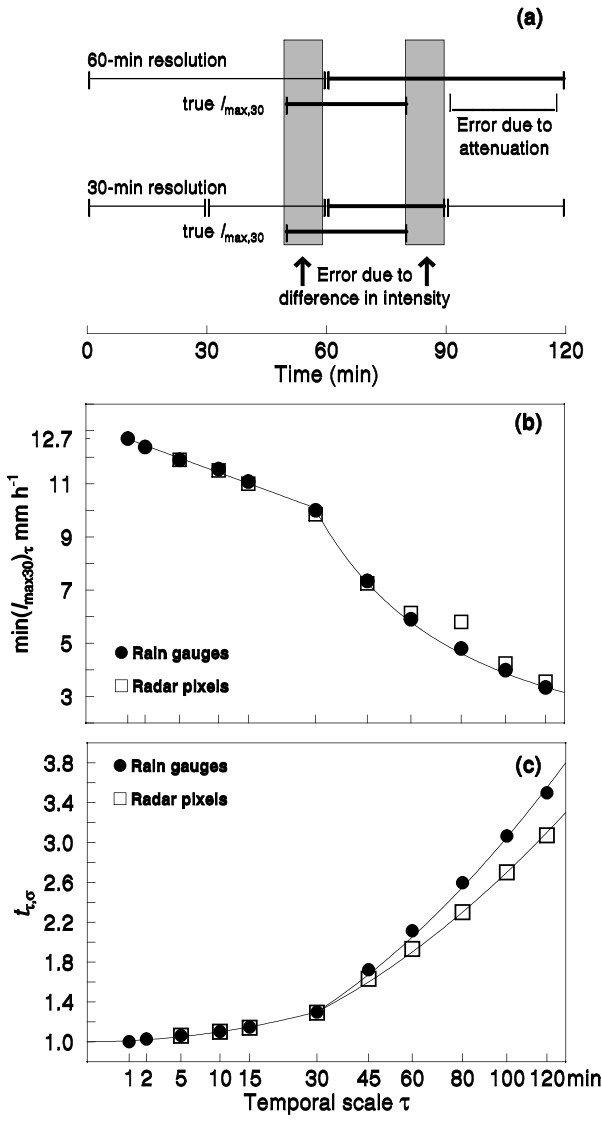

**Figure 2: a)** Time periods influencing the underestimation of $I_{max30}$ when temporal resolution is 30 min (or more) or when temporal resolution is 60 min (or any resolution >30 min). **b)** Minimum threshold for $I_{max30}$ ($\min(I_{max30})_\tau$) derived from rain gauge (solid circles) and radar data (open squares) required to obtain the same number of erosive events as with a temporal resolution of 1 min; lines show Eq. (6.1) and Eq. (6.2) (RMSE is 0.10 and 0.39). **c)** Scaling factor $t_{\tau,\sigma}$ to scale $R_e$ or $R$ for temporal resolution $\tau$ when spatial resolution $\sigma$ is either rain gauge scale (solid circles) or 1 x 1 km² (open squares) respectively; lines show Eqs. (7.1), (7.2) and (7.3) (for all RMSE ≤ 0.04). The x-axes in b) and c) are square-root scaled.




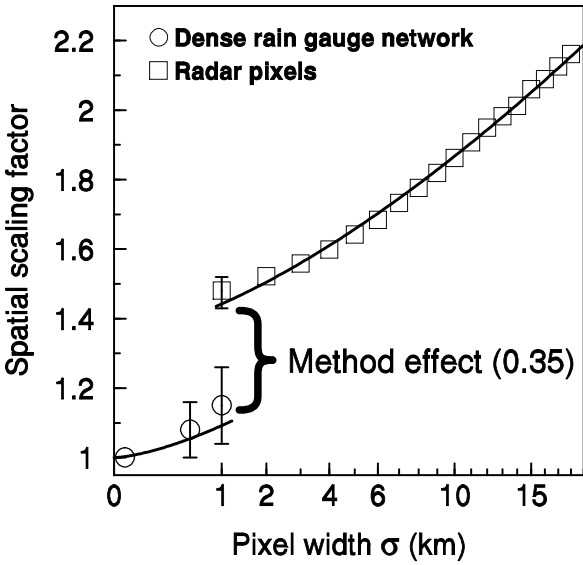

**Figure 3: Spatial scaling factors for *R*. Open circles result from rain gauges aggregated to pseudo-radar pixels. Open squares result from radar and aggregation of radar data. Error bars represent the 95% confidence interval. Lines denote a multiple regression (see text). The x-axis is square-root scaled to improve visibility at low pixel width.**

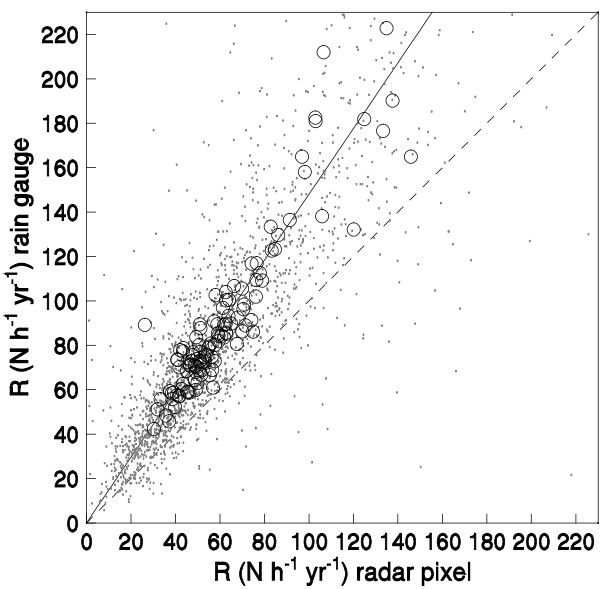

**Figure 4: Annual erosivity $R_y$ (grey points) and multi-annual mean erosivity *R* (black circles) derived from radar pixel and rain gauge data for 115 point-pixel pairs and 16 years. The difference in slope between the solid line and unity (dashed line) is due to the spatial scale and the method effects.**



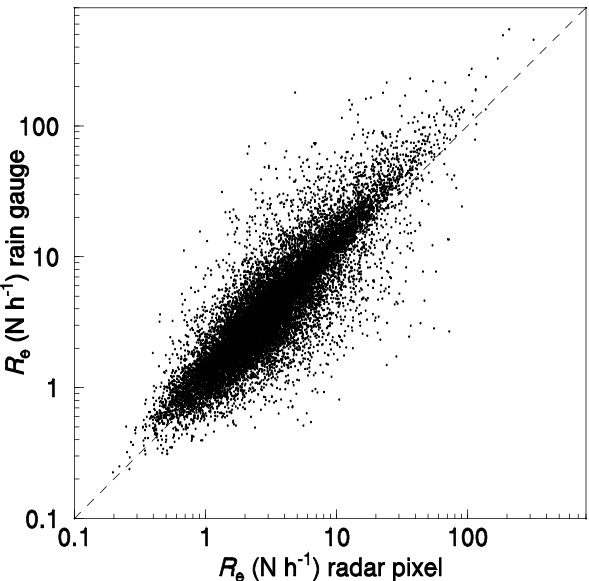

**Figure 5: Comparison of event erosivity $R_e$ calculated from radar data and $R_e$ from rain gauge data for 115 radar pixels that enclose a rain gauge. Only events that were erosive at both scales (19 944 events) during the 16 year period are shown. The dashed line represents unity. Axes are log scaled. Note: no spatial scaling factor or method factor was applied because these factors also**
5    **included the effect of incomplete coverage of the pixel by an erosive rain cell.**

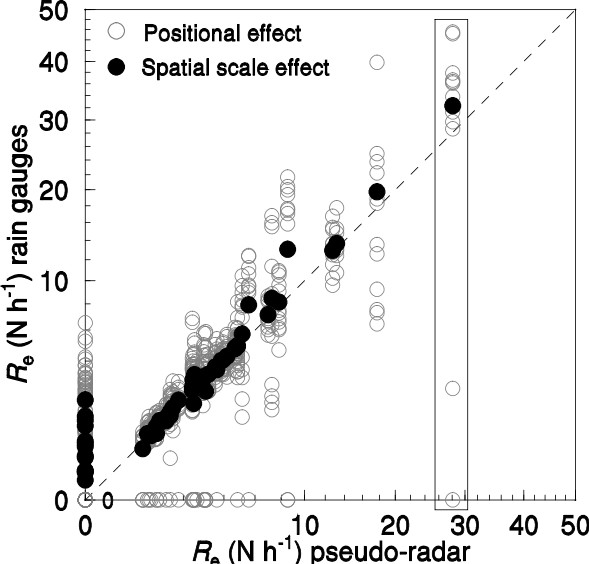

**Figure 6: Event erosivity $R_e$ at 12 rain gauges located within a 1 km² pixel versus $R_e$ based on pseudo-radar data calculated from the hyetographs of the 12 rain gauges (open grey circles). Filled black circles show the average $R_e$ of all 12 rain gauges vs the $R_e$ from pseudo-radar rainfall. Note that the average $R_e$ can be considerably larger than zero while the averaged rainfall of the**
10    **pseudo-radar remains below the thresholds of erosivity (black circles along the y axis). Rectangular frame shows variation of $R_e$ for a single day. Axes are square-root scaled to improve resolution at low $R_e$.**



**Table 1: Overview of the data used to determine the positional effect, the spatial scale effect, temporal scale effect and the method effect.**

| Purpose | Measurement | Spatial scale | Temporal scale | Number of stations/ pixels | Period | Event number |
|---|---|---|---|---|---|---|
| Positional and spatial scale | Rain gauge | Point | 60 min | 115 | 16 yr | 29 610 |
| | Radar | 1 km² | 60 min | 115 | 16 yr | 25 884 |
| Spatial scale and method effect | Rain gauge | Point | 1 min | 12 | 4 yr, Apr - Oct | 542 |
| | Radar | 1 km² | 60 min | 480 x 10$^3$ | 2 months | 1.9 x 10$^6$ |
| Temporal scale | Rain gauge | Point | 1 min | 17 | 16 yr | 4 599 |
| | Radar | 1 km² | 5 min | 17 | 16 yr | 3 924 |

5  **Table 2: Percentage of cases that were erosive at point (115 rain gauges) or at pixel scale (115 radar pixel) relative to a total of 35 124 point-pixel pairs of rain events that were erosive on at least one of both scales.**

| Point scale | Pixel scale | Percentage |
|---|---|---|
| Erosive | Not erosive | 27% |
| Not erosive | Erosive | 16% |
| Erosive | Erosive | 57% |

**Table 3: Percentage of cases that were erosive at point (rain gauge) or at pixel scale, using the pseudo-radar data; in total 579 point-pixel pairs of rain events were erosive on at least one of both scales.**

| Point scale | Pixel scale | Percentage |
|---|---|---|
| Erosive | Not erosive | 9% |
| Not erosive | Erosive | 6% |
| Erosive | Erosive | 85% |

