# Peer review of "Temporal and spatial scale and positional effects on rain erosivity derived from point-scale and contiguous rain data"

_Hydrology and Earth System Sciences, 2018_

## Short Comment (SC1) · 22 Jun 2018

1) As per the authors, the threshold value that determines whether or not a rainfall event is erosive is 12.7 mm or 12.7 mm/hr (see P-3 LN-30). As per the definition of the threshold value on rainfall intensity, the duration of the rainfall intensity is 30 minutes (see P-3 LN-29). Consider a rainfall event that lasted for 30 minutes and contributed 6.4 mm. Would this rainfall event be erosive? As per the definition, the rainfall event contributes 6.8mm. Therefore, the rainfall event is not erosive. However, the intensity of the rainfall event is 12.8mm/hr (=6.4/0.5=6.4*2). Therefore, the rainfall event is erosive. Is this what meant by the definition?

[Figure]

2) As per the authors, the threshold value that determines whether or not a rainfall event is erosive is 12.7 mm or 12.7 mm/hr (see P-3 LN-30). From the reader's point of view, since this threshold value may not be constant in spatial, this threshold value needs to be assessed against the local (i.e., Germany) climate/soil/landuse conditions. As far as I remember, in the past, an attempt was made by a SWAT research group on a project funded by USDA/EPA to establish a global matrix of this threshold value (i.e., threshold values for the UN nations). The research group was trying to couple its methodology with the available soil/LULC/climate databases. Though I am not sure the current status of the project and what has been materialized, it would be worth to check the threshold value that is applicable in Germany.

3) As per the authors, erosivity is the product of a rainfall event's maximum 30-min intensity and its kinetic energy (See P-3 LN-29; P-2 LN-31). As per the authors, both factors depend on rain intensity and thus, intensity is squared in erosivity (see P-2 LN-32).Consequently, the authors state that a difference in rain intensity of 10% would result in difference in erosivity of 21 %( see P-2 LN-33). How did you end up with 21%? Probably, it would be more appropriate to show an example. Consider a rainfall event that lasted for 30 minutes and contributed 43 mm. For this rainfall event, the rainfall intensity is 86 mm/hr (=43/0.5). As per your equation (1) and equation (2.3), the erosivity is 28.33*(43)*(43/0.5)=104.764 unit. If the rainfall intensity is decreased by 10%, as per your equation (1) and equation (2.3), the erosivity is 28.33*(43*0.9)*(43*0.9/0.5) =84.859 unit. Therefore, the difference is 19% (= (104.764-84.859)/ 104.764*100). Moreover, why would you say that the rainfall intensity is squared in erosivity? Is it based on your equations (2.1-2.3)? Would it be incorrect if we conclude that the equations (2.2-2.3) are independent of rainfall intensity? Don't those equations (i.e., 2.2. and 2.3) lead to a constant value regardless of the magnitude of the rainfall intensity?

4) As per the authors, the erosivity of a rainfall event is defined and controlled by the rainfall intensity (see P-3 LN-28). Even though rainfall intensity is the main factor, there are also other striking factors such as angle of rainfall attack, land use, soil structure/texture, and more importantly prior rainfall events, which influence the erosivity. For example, a location that received 8 mm/hr may not be considered eroded as it has not met the threshold (i.e., 12.7 mm/hr).However, at the same location, the next rainfall event with a value of 6mm/hr may detach and erode the soil particles due to fact that the prior rainfall event (i.e., 8 mm/hr) may have already detached and loosened the grains.

5) Consider a location that has received two continuous rainfall events. Assume that the first rainfall event meets your threshold value (i.e., 12.7 mm/hr).In other words, as per your definition, the first rainfall event is considered erosive. Now, assume that the rainfall intensity associated with the second rainfall event is 11mm/hr. Do you think that it is really needed to have 12.7mm/hr to detach the soil that has already been exposed and detached by the first rainfall event that is considered erosive as per your definition? Wouldn't it be needed to have a lesser rainfall intensity to detach the second/exposed/detached soil layer?

6) In equations (2.1-2.3), aren't the upper (76.2 mm/hr) and lower (0.05 mm/hr) threshold values spatial sensitive. Moreover, with the definition presented in the current version of the manuscript, regardless of the energy that is computed using equations (2.1-2.3), a rainfall event is considered non-erosive if the rainfall intensity (i.e., maximum 30-min rainfall intensity) is less than 12.7 mm/hr. Considering this definition, in equation (2.1), what is the reason to set the lower threshold value to 0.05mm. Why would it be required to consider the kinetic energies induced by the rainfall intensities that are less than 12.7 mm/hr? Shouldn't the lower threshold value be 12.7 mm/hr?

---

## Author Comment (AC1) · 11 Jul 2018

We thank S. Mylevaganam for his interest in our work and his rapid comments. In our reply we refer to his numbering of comments.

Reply 1) We did not define the thresholds but they were defined by Wischmeier (1959, but see also Wischmeier and Smith 1958, 1978). According to Wischmeier, a rainfall is regarded erosive if at least one of both criteria meets the requirements. A rain with 6.4 mm in 30 min is thus regarded erosive but its erosivity will be very low (about 1 permil of our highest erosivity).

[Figure]

Reply 2) Mylevaganam (2018) is correct that it is conceivable that these thresholds may vary locally (soils will vary even within one slope) and even temporally (e.g. depending on antecedent soil moisture). Indeed, different thresholds have been proposed for Germany (10 mm or 10 mm/hr; Rogler and Schwertmann 1981) but these thresholds have never been tested and confirmed. We are aware of two studies, one in Germany (Martin 1991) and one in Nigeria (Sabel-Koschella 1988), which tried to verify the thresholds and the duration over which the maximum intensity has to be determined but without any improvement. The simple reason is that erosivity close to the threshold is very low and does not exceed uncertainty of the erosion measurement, which is large. Also uncertainty of rain (gauge) measurement is in the range of 1 to 2 mm (wind drift, adhesion, mechanical resistance etc.).

The SWAT model does not use rain erosivity because it is a landscape model that includes the change in transport capacity along the flow path into the calculation. It thus depends on the so-called initial abstraction, which acts similarly as the thresholds of rain erosivity although the initial abstraction basically applies to runoff and not to erosion. Many attempts have been made to regionalize initial abstraction.

Reply 3) 121% result from squaring 110%. At this part of the manuscript the equations have not been introduced and hence the reader cannot know that the calculation of erosivity is more complicated. Exactly 121% would only result for conditions leading to eqn 2.3 throughout the entire rain event while other values would be obtained for condition requiring eqn 2.1 or eqn 2.2. The squaring results from the fact that eqn 2.1 to 2.3 only calculate the kinetic energy per mm of rain and hence the result of these equations have to be multiplied with the amount of rain per time increment and summed up . This sum has then to be multiplied with the maximum 30-min intensity, which causes the squaring (Wischmeier 1959; Wischmeier and Smith 1958, 1978). Hence with all three equations intensity has to be squared and our sentence and the simple example to illustrate the problem of non-linearity in the introduction is still correct.

Reply 4) This is true and well known. For individual events especially wind speed and

drop size distribution my deviate considerably from the average conditions (e.g. see Brandt 1990, Iserloh et al. 2013). However, the necessary data are usually not available because drop size is usually not measured and wind speed during thunderstorms varies locally. Furthermore, to the best of our knowledge, no algorithm exists that would incorporate this information in erosivity calculation. Other influences like land use and soil structure/texture have no impact on rain erosivity but they influence the resistance against erosivity. These influences are considered in other factors of the Universal Soil Loss Equation.

A sequence of rains as proposed Mylevaganam (2018) would meet the definition of an erosive event if both rains would be separated by not more than 6 hr. For a longer separation we can expect that the soil stabilizes and gains its resistance against raindrops again (e.g. due to drying and the attractive forces of water menisci).

Reply 5) It may be that a lower threshold would apply for the following rain but also the opposite could be true and thresholds could increase for a subsequent rain because of the solid seal developed during the preceding rain. These influences are beyond the concept of rain erosivity as an entirely rain dependent property without consideration of soil or crop management that are entered at a later point of erosion calculation (Wischmeier and Smith 1978). This concept of rain erosivity explained from 72 to 97% of the variation in individual-storm erosion from tilled continuous fallow (Wischmeier 1959) and it is in use since then. In particular this concept was able to explain soil loss of more than 10'000 plot yr sufficiently accurate to serve as sound basis for conservation farm planning (Wischmeier and Smith 1978).

Reply 6) The simple reason is that rain intensity is never constant during a rain and periods with an intensity of 0.05 mm/hr or lower can also occur during a rain that exceeds the criteria of an erosive event. A detailed justification of the equations can be taken from the original publications (Wischmeier 1959, Wischmeier and Smith 1958).

References: Brandt, J.: Simulation of the size distribution and erosivity of raindrops

and throughfall drops, Earth Surface Processes and Landforms 15, 687-698, 1990.

Iserloh, T., Fister, W., Marzen, M., Seeger, M., Kuhn, N. J. and Ries J. B.: The role of wind-driven rain for soil erosion - An experimental approach. Zeitschrift fur Geomorphologie 57(1 SUPPL. 1), 193-201, 2013.

Fischer, F.K., Winterrath, T. and Auerswald, K.: Temporal and spatial scale and positional effects on rain erosivity derived from contiguous rain data, Hydrol. Earth Syst. Sci. Discuss., https://doi.org/10.5194/hess-2018-305, 2018

Martin, W.: Die Erodierbarkeit von Böden unter simulierten und natürlichen Regen und ihre Abhängigkeit von Bodeneigenschaften, Diss., TU München. 1988.

Mylevaganam S.: Interactive comment on "Temporal and spatial scale and positional effects on rain erosivity derived from contiguous rain data" by F. K. Fischer et al., Hydrol. Earth Syst. Sci. Discuss., https://doi.org/10.5194/hess-2018-305-SC1, 2018.

Rogler, H., and Schwertmann, U.: Erosivität der Niederschläge und Isoerodentkarte Bayerns, Journal of Rural Engineering and Development, 22, 99–112, ISSN 0044-2984, 1981.

Sabel-Koschella, U.: Field studies on soil erosion in the southern guinea savanna of western Nigeria Diss., TU München, 1988.

Wischmeier, W. H.: A rainfall erosion index for a universal soil-loss equation, Soil Sci. Soc. Am. Pro., 23, 246–249, 1959.

Wischmeier, W. H., and Smith, D. D.: Rainfall energy and its relationship to soil loss, T. Am. Geophys. Un., 39, 285-291, 1958.

Wischmeier, W. H., and Smith, D. D.: Predicting rainfall erosion losses – a guide to conservation planning, U.S. Department of Agriculture, Agriculture Handbook No. 537, Washington, DC, 1978.

---

## Referee Comment (RC1) · Anonymous Referee #1 · 15 Aug 2018

Excellent paper on very interesting and actual topic. There is wide discussion about application of various rain data sources for determination of rain erosivity for application within USLE, but there are very few papers, dealing with this topic on relevant level. And even less information about possible corrections and expected errors and problems. What I appreciate a lot is data set size – number of stations, area included and duration of the study (number of events recorded and included). I have no comments or requests to change or add anything from scientific point of view – on this point I strongly recommend for publication.

I only have several minor comments to formal presentation of the paper – to be possibly

more clear to the readers or/and easily understandable – as such statistic studies are always difficult to interpret to someone, who did not study the certain problem deeply. Introduction: potential recent data sources are well discussed – (gauging stations networks and meteo-radars) also including their accuracy. To be fair, I would appreciate also short discussion of accuracy and potential errors occurring on gauging stations. There are for sure errors in records, especially during extreme stormy events given by tipping bucket, by capacity of drainage pipe (if this type of gauging station is used), etc. It also depends a lot on type of device used. Also, there is modern recent method now for rainfall parameters estimation using commercial microwave links. I fully understand that these data are not analyzed within this paper, but they should at least be mentioned in Introduction part. Hypothesis formulation are relevant and clear. They are relatively trivial – and expectable – therefore I would appreciate possibly to more clearly state if those are research questions, which shall be answered in Conclusions and Discussion. Chapter 2 – to be clearer, I would recommend to characterize at least briefly goal and basic scheme of analyses planned (done) of the research in the beginning of the chapter. It is then described later – but reader is a bit confused by overview of methodology, but not knowing, which data will then be used and why actually. Chapter 2.2, section 15 – there is a bit confusing for me discrepancy between 16 years (duration of whole experiment = data record ?) and four years for 12 rainfall gauging stations within 1 km2. Can be explained better ? Basic description of gauging stations (equipment) and analyzed data shall be performed to clarify number of rising associated questions – from both of gauging stations and from radars. Were rainfall data from gauging stations treated, corrected, filled gaps, …. ? Time resolution and other data characteristics, … basic statistics of the data set should be performed (really all the stations measured all the time for whole 16 years ?). Is there consistency in equipment ? (=all the stations had same equipment during whole period ?) Figure 1 – relation between sections B and C is not really clearly described. Why Thiessen polygons were used and not some smooth interpolation polygons ?

Generally – all my recommendations are just minor in importance and formal to clarify

the analyses performed and I appreciate the paper as a whole a lot.

---

## Author Comment (AC2) · 10 Sep 2018

We appreciate the encouraging comments by the referee.

Regarding his suggestion to mention microwave links in the Introduction we added to the Discussion.

"...The same is true for using data of commercial microwave links, which recently have been identified as additional source for retrieving precipitation (Chwala et al., 2012; Overeem et al., 2013) and which will require the method effect to be adapted for this particular approach. The approach is based on analysing the signal attenuation that

depends on rain intensity. These data are especially valuable in regions with sparse coverage by conventional measurement devices like, e.g., in parts of the African continent, but may also improve high resolution precipitation estimates and forecasts in hydrometeorological applications (Chwala et al., 2016)."

Regarding accuracy and potential errors at gauging stations we added to the new Chapter 2.1 Data sets:

"Precipitation measurements of the DWD station network were conducted with Pluvio Ott weighing rain gauges (OTT Hydromet GmbH, Kempten, Germany) with a collector area of 200 cm2, a measurement range of 0-1800 mm/h, and a 1-minute resolution of 0.1 mm/h. The precipitation data passed a quality control system testing for completeness, carrying out climatological tests, checking consistency over time as well as internal and spatial consistency (Spengler, 2002; Kaspar, 2013). The data were neither corrected for wind drift effects nor homogenized. A thorough overview of the precision of rain gauge measurements is given in Monesi et al (2009). Information on the stations' meta data can be found in the Climate Data Center (ftp://ftp-cdc.dwd.de/pub/CDC/observations_germany/climate/hourly/precipitation/historical/) of DWD."

We also expanded the description of the radar data by adding:

"The DWD radar network underwent several upgrades during the analysis period. In the beginning of the considered time period five single-polarization systems (DWSR-88C, AeroBase Group Inc., Manassas, USA) operated without Doppler filter the latter being added between 2001 and 2004. Between 2009 and today, DWD exchanged the network of C-band single-polarization systems of the next generation of type ME-TEOR 360 AC (Gematronik, Neuss, Germany) and DWSR-2501 (Enterprise Electronics Corporation, Enterprise, USA) by modern dual-polarization C-band systems of type DWSR-5001C/SDP-CE (Enterprise Electronics Corporation), all equipped with Doppler filter. During the time of exchange, a portable interim radar system of type DWSR-

5001C was installed at some of sites. Radar data underwent an operational quality control system. They were adjusted to gauge data within a reprocessing suite applying a consistent software version (version 2017.002) and optimized quality control algorithms (Winterrath et al., 2017)."

Regarding the request to give an overview of the data we added at the beginning of chapter 2 a sup-chapter "2.1 Data sets" in which we describe the data and for which question we will use the data. We removed the respective information from the following chapters in order to avoid repetition and increase in manuscript length.

This rearrangement should also have made clearer now that the 16-yr data set and the 4-yr data set are independent data sets. The long-term data were taken from a long-term observation network while the 4-yr data of high spatial resolution (12 recording rain gauges within 1 km$^2$) stem from a research project that did not last longer. Globally, there are hardly any other rain gauge data of similar density available.

Regarding the description (and justification) of Thiessen polygons in Fig. 1, we now added: "A previous geostatistical analysis of the spatial pattern had shown that erosive rains recorded by the dense network followed near-linear trends between neighboring rain gauges (Fiener and Auerswald 2009; see also Fig. 1b for an example). From this follows that the spatial pattern can be retrieved best by linear interpolation between the rain gauge sites. The spatial average of a linear interpolation is mathematically identical to the well-known Thiessen polygons. We thus used Thiessen polygons for calculation of the spatial average because they are mathematically simpler as they lead to a constant weighting for the different stations irrespective of the recorded amount of rain. They also can easily be illustrated (Fig. 1c)."

A point-to-point reply can be found in the supplement.

Please also note the supplement to this comment:
https://www.hydrol-earth-syst-sci-discuss.net/hess-2018-305/hess-2018-305-AC2-

supplement.pdf

[Figure]

**Supplement:**

Excellent paper on very interesting and actual topic. There is wide discussion about application of various rain data sources for determination of rain erosivity for application within USLE, but there are very few papers, dealing with this topic on relevant level. And even less information about possible corrections and expected errors and problems. What I appreciate a lot is data set size – number of stations, area included and duration of the study (number of events recorded and included). I have no comments or requests to change or add anything from scientific point of view – on this point I strongly recommend for publication.

We appreciate the encouraging comments

I only have several minor comments to formal presentation of the paper – to be possibly more clear to the readers or/and easily understandable – as such statistic studies are always difficult to interpret to someone, who did not study the certain problem deeply.

Introduction: potential recent data sources are well discussed – (gauging stations networks and meteo-radars) also including their accuracy.

To be fair, I would appreciate also short discussion of accuracy and potential errors occurring on gauging stations. There are for sure errors in records, especially during extreme stormy events given by tipping bucket, by capacity of drainage pipe (if this type of gauging station is used), etc. It also depends a lot on type of device used. Also, there is modern recent method now for rainfall parameters estimation using commercial microwave links. I fully understand that these data are not analyzed within this paper, but they should at least be mentioned in Introduction part.

We added reference to commercial microwave links in the discussion (not in the introduction).

"…The same is true for using data of commercial microwave links, which recently have been identified as additional source for retrieving precipitation (Chwala et al., 2012; Overeem et al., 2013) and which will require the method effect to be adapted for this particular approach. The approach is based on analysing the signal attenuation that depends on rain intensity. These data are especially valuable in regions with sparse coverage by conventional measurement devices like, e.g., in parts of the African continent, but may also improve high resolution precipitation estimates and forecasts in hydrometeorological applications (Chwala et al., 2016)."

Regarding accuracy and potential errors at gauging stations we added to the new Chapter 2.1 Data sets:

"Precipitation measurements of the DWD station network were conducted with Pluvio Ott weighing rain gauges (OTT Hydromet GmbH, Kempten, Germany) with a collector area of 200 cm$^2$, a measurement range of 0-1800 mm/h, and a 1-minute resolution of 0.1 mm/h. The precipitation data passed a quality control system testing for completeness, carrying out climatological tests, checking consistency over time as well as internal and spatial consistency (Spengler, 2002; Kaspar, 2013). The data were neither corrected for wind drift effects nor homogenized. A thorough overview of the precision of rain gauge measurements

is given in Monesi et al (2009). Information on the stations' meta data can be found in the Climate Data Center (ftp://ftp-cdc.dwd.de/pub/CDC/observations_germany/climate/hourly/precipitation/historical/) of DWD."

We also expanded the description of the radar data by adding:

"The DWD radar network underwent several upgrades during the analysis period. In the beginning of the considered time period five single-polarization systems (DWSR-88C, AeroBase Group Inc., Manassas, USA) operated without Doppler filter the latter being added between 2001 and 2004. Between 2009 and today, DWD exchanged the network of C-band single-polarization systems of the next generation of type METEOR 360 AC (Gematronik, Neuss, Germany)  and DWSR-2501 (Enterprise Electronics Corporation, Enterprise, USA) by modern dual-polarization C-band systems of type DWSR-5001C/SDP-CE (Enterprise Electronics Corporation), all equipped with Doppler filter. During the time of exchange, a portable interim radar system of type DWSR-5001C was installed at some of sites. Radar data underwent an operational quality control system. They were adjusted to gauge data within a reprocessing suite applying a consistent software version (version 2017.002) and optimized quality control algorithms (Winterrath et al., 2017)."

Hypothesis formulation are relevant and clear. They are relatively trivial – and expectable – therefore I would appreciate possibly to more clearly state if those are research questions, which shall be answered in Conclusions and Discussion.

We added at the bottom of the Introduction:
"We will quantify these effects and discuss their implications."

Chapter 2 – to be clearer, I would recommend to characterize at least briefly goal and basic scheme of analyses planned (done) of the research in the beginning of the chapter. It is then described later – but reader is a bit confused by overview of methodology, but not knowing, which data will then be used and why actually.

We added at the beginning of chapter 2 a sup-chapter "2.1 Data sets" in which we describe the data and for which question we will use the data. We removed the respective information from the following chapters in order to avoid repetition and increase in manuscript length.

Chapter 2.2, section 15 – there is a bit confusing for me discrepancy between 16 years (duration of whole experiment = data record ?) and four years for 12 rainfall gauging stations within 1 km2. Can be explained better ?

Due to the rearrangement of information in a sub-chapter "2.1 Data sets" it should be clearer now that these are independent data sets. The long-term data were taken from a long-term observation network while the 4-yr data of high spatial resolution (12 recording rain gauges within 1 km²) stem from a research project that did not last longer. Globally, there are hardly any other rain gauge data of similar density available.

Basic description of gauging stations (equipment) and analyzed data shall be performed to clarify number of rising associated questions – from both of gauging stations and from radars. Were rainfall data from gauging stations treated, corrected, filled gaps,…. ? Time resolution and other data characteristics, …basic statistics of the data set should be performed (really all the stations measured all the time for whole 16 years ?). Is there consistency in equipment ? (=all the stations had same equipment during whole period ?)

We added an extensive description to the new Chapter 2.1 Data sets (see above)

Figure 1 – relation between sections B and C is not really clearly described. Why Thiessen polygons were used and not some smooth interpolation polygons ?

We added:
"A previous geostatistical analysis of the spatial pattern had shown that erosive rains recorded by the dense network followed near-linear trends between neighboring rain gauges (Fiener and Auerswald 2009; see also Fig. 1b for an example). From this follows that the spatial pattern can be retrieved best by linear interpolation between the rain gauge sites. The spatial average of a linear interpolation is mathematically identical to the well-known Thiessen polygons. We thus used Thiessen polygons for calculation of the spatial average because they are mathematically simpler as they lead to a constant weighting for the different stations irrespective of the recorded amount of rain. They also can easily be illustrated (Fig. 1c)."

Generally – all my recommendations are just minor in importance and formal to clarify the analyses performed and I appreciate the paper as a whole a lot.

---

## Referee Comment (RC2) · Anonymous Referee #2 · 7 Nov 2018

Due to unknown mechanism of cloud microphysics or cloud dynamics, it is supposed that rainfall prediction and radar rain calibration in detailed small space and time scale resolution such as certain 100 m2 is unreliable, even with the technology in the state of arts. Also, the space resolution is influenced by spatial fluctuation of soil surface property, topography, geology and geo-structure. It can vary widely even in 10 m2 scale. As you know, only average values of rainfall or erosion in the limited resolution are available in real condition. In this situation, I recommend the research rather focused on the minimum threshold which time and space resolution is suitable for clarify the positional effects on rain erosivity.

---

## Author Comment (AC3) · 7 Nov 2018

We thank Ref. #2 for his efforts in reviewing our manuscript.

We fully agree with this comment that radar technology does not perfectly resolve precipitation on small temporal and spatial scales. Nevertheless it is important to close the gap between the point data at rain gauges and the spatial scale provided by radar (or the even larger scale by satellite data; see Vrieling et al., 2010, 2014). This is why we used a high-density rain gauge field to include smaller scales.

For the application neither the point scale nor the radar or satellite scale is usually

of interest but this may be plots, fields, or catchments. This means that a user has to decide which data are closest to his scale of interest and he has to close the gap between both scales. Our analysis will guide this decision and provide relations to close the gap. Also the importance of the positional effect strongly depends on the research question and the study area. Importance increases the shorter the time span under focus becomes and the more convective rains prevail in the study area.

Vrieling, A., Sterk, G., de Jong, S.M.: Satellite-based estimation of rainfall erosivity for Africa. J. Hydrol., 395, 235-241, 2010.

Vrieling, A., Hoedjes, J.C.B., van der Velde, M.: Towards large-scale monitoring of soil erosion in Africa: Accounting for the dynamics of rainfall erosivity. Global Planetary Change, 115, 33-43, 2014.